# Aggressive strategies of the COVID-19 pandemic on the apparel industry of Sri Lanka using structural equation modeling

Wasantha Rajapakshe[ID]*, D. S. M. Karunaratna, W. H. G. Ariyaratne, H. A. Lakshani Madushika, G. S. K. Perera, P. Shamila[ID]

Department of Business Management, Sri Lanka Institute of Information Technology, SLIIT Business School, Malabe, Sri Lanka

* wasantha.r@sliit.lk

**Data Availability Statement:** The data underlying the results cannot be shared publicly because of data protection due to ongoing analysis but are

## Abstract

During the COVID-19 crisis, the apparel industry faced many challenges. Aggressive cost-cutting strategies became a top priority, and in turn, these influenced stressors and adversely affected business sustainability. This study examines the impact of aggressive strategies during the COVID-19 pandemic on business sustainability in the apparel industry of Sri Lanka. Further, it investigates whether the relationship between aggressive cost-cutting strategies and business sustainability was mediated by employee stress, considering aggressive cost reduction strategies and workplace environmental changes. This was a cross-sectional study with data collected from 384 employees in the apparel industry in Sri Lanka. Structural Equation Modelling (SEM) was applied to analyze the direct and indirect effects of aggressive cost reduction strategies and workplace environmental changes on sustainability with mediating effects of stress. Aggressive cost reduction strategies (Beta = 1.317, p = 0.000) and environmental changes (Beta = 0.251, p = 0.000) led to an increase in employee stress but did not affect business sustainability. Thus, employee stress (Beta = -0.028, p = 0.594) was not a mediator in the relationship between aggressive cost-cutting strategies and business sustainability; business sustainability was not a dependent variable. The findings proved that managing workplace stress, particularly improving stressful working environments and aggressive cost reduction strategies, can enhance employee satisfaction. Thus, managing employee stress could be beneficial for policymakers to focus on the area(s) required to retain competent employees. Moreover, aggressive strategies are unsuitable to apply during crisis to enhance business sustainability. The findings provide additional knowledge to the existing literature, enabling employees and employers to predict causes of stress and serve as a significant knowledge base for further studies.

## Introduction

Managing and coordinating environmental, social, and financial demands and concerns to achieve responsible, ethical, and long-term success are known as business sustainability [1].

available from the corresponding author on reasonable request. Further, data are available from the Sri Lanka Institute of Information Technology, SLIIT Business School, Sri Lanka (contact via Prof. Samantha Thelijjagoda, Dean, SLIIT Business School, Sri Lanka Institute of Information Technology, New Kandy Road, Malabe, Sri Lanka. Email: samantha.t@sliit.lk) for researchers who meet the criteria for access to confidential data.

**Funding:** The author(s) received no specific funding for this work.

**Competing interests:** The authors have declared that no competing interests exist.

(Dumitrescu, et al., 2013)Sustainable business strategies bring significant financial and environmental benefits to manufacturers, such as cost-effective procedures that reduce negative environmental consequences while preserving energy and natural resources. Employees, community, and product safety too are improved by sustainable production [2]. Three key components can be identified regarding business sustainability: employee satisfaction (people), long-term sustainable profit (profit), and a more sustainable planet with fewer resources polluting the environment (planet) [3].

To increase growth and worldwide competitiveness, many firms include "sustainability" as a key goal in their strategy and operations [4]. According to the World Economic Forum, sustainability is the leading priority on the global agenda [5]. It has become the main factor in executive decision-making and a key topic in discussions among top management [6]. In 2019, McKinsey listed sustainability as one of the apparel industry's top priorities [7]. Developing raw material sourcing, better recycling processes, decreasing water consumption, and substituting hazardous chemicals with harmless alternatives are among the measures to cut the fashion industry's environmental footprint and assist the industry in surviving amid demanding and volatile business situations [6,7].

Greenhouse gases (GHGs) absorb solar energy and keep heat trapped and close to the earth's surface, significantly impacting climate change. The fashion industry accounts for 10% of global carbon emissions, while dyeing and the finishing stages significantly contribute to the apparel industry's carbon emissions [8]. One of the largest apparel manufacturers in Sri Lanka, the Hirdaramani Group has achieved net-zero GHG emissions from energy in all its Sri Lankan factories [9]. Further, garment manufacturing generates considerable textile waste alongside the supply chain. A study on waste generation in the garment industry in Sri Lanka proves that a considerable 39.4% of the total wastes, followed by fabric and yarn, is around 29.6% [10]. The procedures adopted/followed in Sri Lanka's apparel manufacturing facilities comply with worldwide environmental norms and requirements [11]. Sri Lanka is already leading the way regarding environmental sustainability in the garment business while inspiring the rest of the globe to address the industrialization-related environmental crisis [6,7,11].

In 2020, the outbreak and the rapid spread of the global Covid-19 pandemic was an unexpected incident with enormous magnitude, resulting in economic impacts in addition to health concerns. International trade declined, supply chain operations were interrupted, and orders were canceled due to rapidly declining demand, showing the vulnerability of global supply chains to the COVID-19 pandemic. Unlike in earlier crises such as the severe acute respiratory syndrome coronavirus (SARS-CoV) and the Middle East respiratory syndrome coronavirus (MERS-CoV), during COVID-19, all activities across the value chain almost collapsed adversely affecting its up- and down-streams, posing a threat to apparel/ raw material suppliers (with disruptions to operations/raw material availability), employees/workers (with job cuts), suppliers, brands/retailers/buyers, and consumers [12–14]. This downturn in the business environment prompted every player/stakeholder in the industry to rethink their strategies and goals [7,15]. Due to this global pandemic, the consumption of America lost 60–65% of consumer disruptions and 25% of Canada and the EU [16]. According to the latest investigation of Sustainable Apparel Coalition members, one-third (1/3) of decision-makers in the fashion brands, sellers, and manufacturers perceived that they needed more time to be ready for the crisis of Covid-19 [17]. These findings explain how significant it was for businesses to adopt coping strategies for survival. Most companies, including the apparel industry, were pushed to resort to aggressive cost reduction among their belt-tightening strategies, battling the pandemic's squeeze on bottom lines.

The COVID-19 pandemic also devasted the Sri Lankan apparel industry, one of the significant contributors to the country's economy. In 2019, the industry generated US$ 5.2 billion in

export revenue, accounting for 44% of the country's total exports. It is uncertain to ascertain the damage beyond this level, as market contractions fell, Sri Lanka's apparel exports almost halved from US$ 5.2 Bn to US$ 2.9 Bn in 2019–2020 [18]. This circumstance saw that small players failed to survive, whereas big players found it challenging to remain afloat, with salaries accounting for 70% of the total cost [19].

Then, as the COVID-19 second wave emerged, several workers at Brandix, one of Sri Lanka's largest apparel manufacturers, tested positive for COVID-19 in October 2020. The cluster at the Brandix company resulted in 10,000 cases during the next two months, accounting for half of the total cases in Sri Lanka [15]. Employees at Brandix said they were compelled to report to work despite being COVID-19-infected [20]. The Joint Apparel Association Forum (JAAF) in Sri Lanka anticipated that the pandemic affected 100,000 job losses in the apparel sector, with a 40% decrease in clothing and textile exports in the fiscal year 2020 [21].

According to the statistics, more than 70% of fabric and 70%-90% of accessories required for production are imported [22]. Imports were disrupted by lockdowns, emerging COVID-19 waves, and clusters, thus disrupting/delaying production and export shipments. When considering the lackluster state of the global apparel market, Sri Lankan apparel manufacturers saw the urgent need to devise tactics to transform the then-current crisis into a competitive edge in an evolving industry. The overall profit margin/percentage of apparel export orders which enables business sustainability has been heavily influenced by cost management or cost savings approaches necessitated due to the pandemic conditions in apparel production. Many studies emphasize the importance of such a position to embrace sustainability in the apparel sector.

## Problem statement

The COVID-19 second wave in Sri Lanka originated from a Sri Lankan-reputed apparel company. During this pandemic, the Sri Lankan apparel industry has been facing challenges with interrupted supply chains, uncertainties in future demand, cash flow constraints, and recurrent lockdowns imposed occasionally in the country, causing operational disruptions. Moreover, this Covid-19 pandemic increased employee stress levels, which also increased overhead costs. Hence, apparel companies saw the rising cost for employees' health. Because of social distance rules, the factory layout was expected to be maintained with sufficient space (unlike before the pandemic).

As some factories were closed temporarily due to the Covid-19 pandemic, apparel companies switched to alternatives like sub factories/sub orders to carry out their supply chain activities. Therefore, they had to face substantial transportation costs.

This severe pandemic affected the Sri Lankan apparel industry because apparel firms had to meet stringent regulations to maintain good sustainability practices. In that case, the apparel companies have tended to reduce costs during this pandemic, according to the rules and regulations. Many firms in sectors afflicted by the Covid-19 pandemic implemented salary cuts rather than layoffs to preserve their employees for a quick recovery [21]. Waste management, freezing recruitment, and applying lean six sigma were basic tactics used by the firms to manage expenditures [7,11,19]. Furthermore, the workplace environment has drastically changed, prioritizing health and safety. Maintaining social distance, restricting social functions and gatherings, and working from home (WFH) were standard measures adopted to avoid the rapid spread of the Covid-19 virus. After changing/modifying some workplace environments, employees' stress levels increased mainly due to sudden changes to their working habits/lifestyle, like social isolation. Therefore, employees needed a better mentality/emotional support, and their performance highly affected the business's sustainability. Therefore, these factors were highly challenging and impacted maintaining sustainability in the organization. The

pandemic disrupted the revenue channels of businesses, pushing companies to retrench expenses for organizational survival. Resulting in narrowing profit margins, this situation exacerbated the severely affected apparel industry, for which sustainability regulations (by then already in force) are mandatory, among other issues.

This paper focuses on how aggressive cost reduction strategies, workplace environmental changes, and employee stress affect business sustainability in the apparel industry of Sri Lanka during the COVID-19 pandemic. This study develops a framework to support the sustainable development of the apparel industry and suggests practical and research implications to enhance business success for sustainability post-COVID- 19. Thus, this study is expected to add to the existing literature, and the findings can assist key players in the apparel industry in facing similar crises in the future.

## Contribution of the study

- The findings of this study generate several benefits to the academic field and Sri Lankan working environment, which will assist managers and employees in making decisions during crises.

- First, there needs to be more research on understanding how aggressive strategies such as aggressive cost reduction and environmental changes are adopted by organizations during a crisis and identify its impact on business sustainability. This study is among the first to consider aggressive strategies an essential antecedent of business sustainability in Sri Lankan work environments.

- Second, the study assesses the mediating role of employee stress between aggressive strategies and business sustainability. Thus, it explains the mechanism through which aggressive strategy can influence business sustainability.

- Third, this study's literature review is well grounded with a peer-reviewed database; the results, too, are empirically sound, backed by personal experience. Thus, the study will provide empirical evidence to identify and measure the importance of the causal issues related to employee stress and business sustainability during a similar crisis.

- Finally, the conceptual framework of this study facilitates the understanding of determinant factors of business sustainability, mediating employee stress beyond simple explanations. Thus, research on the conceptual model might generate practical information. It will help policymakers set required policies and managers predict causes for the employees' stress and business sustainability during a crisis environment like COVID-19.

## Literature review

**Business sustainability.** 'Sustainable manufacturing' is defined as producing manufactured goods using cost-effective procedures that reduce negative environmental consequences while preserving energy and natural resources. Sustainable production benefits the community, the workers, and the safety of the products [2]. Companies seek sustainability for various reasons: increased profitability, improved financial performance, enhanced productivity, employee performance, employees health and safety, employee satisfaction, consumer satisfaction, improved operational efficiency, and reduced environmental pollution. Moreover, these contribute to building a company's long-term viability and success and taking advantage of regulatory opportunities and limits. For the apparel industry–if supplying for reputed brands / large buyers, sustainability is vital [2,6,7,9,23,24].

**Aggressive cost reduction strategies.** Cost is an essential factor in any industry, as most industries are more focused on profits. Cost reduction strategies are effective methods to increase/enhance the efficiency of operations, reduce costs, raise productivity, and allow for strategic resource reallocation [25]. During the COVID-19 crisis, cost reduction had become a top priority since production and profitability were disrupted, and the bottom line was in turmoil [26].

Cost-cutting has always been a key focus for business owners, but sustainability has also become a top priority in recent years. Many companies have taken steps to go green/implement green business practices "as it will also save money [27] and generate higher profit margins in the long run. However, it is a prevalent misconception that these two objectives are mutually exclusive. The problem with many cost reduction strategies is that these need to be implemented concerning their long-term sustainability; such strategies implemented by most companies have not been far-sighted and hence can only provide temporary financial relief. Failures in cost-cutting strategies negatively impact company infrastructure and culture, leaving businesses struggling long, even after the crisis has ended [28].

Some of the cost-cutting sustainability ideas/concepts may need an initial investment, but these will pay off in the long run. Trying to execute the ideas simultaneously rather than one at a time sounds sensible. This approach possibly helps attain zero waste in each company [27]. Many researchers investigated aggressive cost-cutting strategies during the COVID-19 pandemic. Some of these are **freezing recruitment** [29,30], **employee layoffs** [31,32], **reducing bonuses** [19,32–34], **reducing employee salaries and wages** by 5% to 60% [21,32,35], **waste management** [7,9,27,36,37], a **collaboration between retailers and suppliers** [38–41], and **Lean Six Sigma** [11,42–44].

## Workplace environmental changes

Social obligations can be very much accomplished with flexible working hours. Due to the Covid-19 threat, the traditional working background has changed. Hence, organizations **encourage WFH** [27,33,45,46], **flexible working hours** [47–49], and **delocalized work** to earn a living uncomplicated for employees [50,51].

Investigating the above-mentioned circumstances has allowed companies to change perceptions to create strategies based on varying consumer preferences and behavioral models since it would be valuable for organizational performance after the Covid-19 pandemic. Moreover, customer preference has been moving towards online platforms, while order fulfillment has been completed physically. Therefore, organizations maintain regular sanitization with social distancing, wearing masks, and other special safety precautions because of employee consciousness about sanitation and individual safety measures [6,51]. Many organizations have been using virtual meetings instead of the usual physical meetings in the workplace to ensure social distancing between two or more people. Also, factories in all sectors, including apparels, displayed posters in their workplaces about instructions on healthy tips that employees should follow during this pandemic [1]. Generally, most workplaces were insecure and harmful to employees in developing countries. This indicates positive and /or negative impacts depending on the main physical conditions in the workplace environment. A healthy and secure workplace helps maintain productivity [52,53].

**Stress on COVID-19.** Many business owners and companies have realized that employees are one of their most valuable assets [24]. COVID-19's impact has propelled new sources of financial, social, and physical stress far beyond the norm. Financial issues (81%), job insecurity (77%), fear of getting infected (60%), and social distancing (47%) have been the top stressors due to COVID-19, according to a new MetLife mental health study [54]. These stressors have been long-lasting and widespread; if neglected, these threaten employee well-being [54].

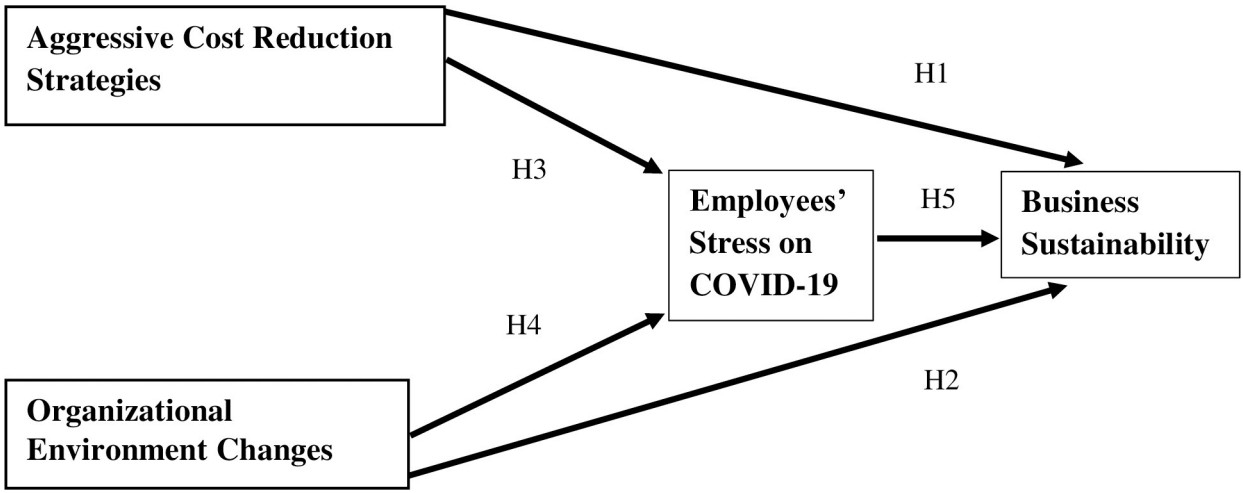

**Fig 1. Conceptual framework.** Source: Author's illustration.

Sri Lanka is ranking fourth /is ranked in fourth place on the 2020 ITUC Global Rights Index, which ranks nations on a scale of 1 to 5+ (worst) for respect for laborers' rights [55]. During the COVID-19 outbreak in Sri Lanka, researchers recorded complaints of the garment worker- rights violations, such as unpaid wages; in factories, union-busting activity, mass unfair dismissals, many employees infected with COVID-19, and migrant laborers trapped in boarding houses or hostels during country lockdowns [21].

Many researchers identify **life threats** [15,56–58], **lockdown** [21,59,60], **isolation and social distancing** [19,61,62], **job insecurity** [31,34,63], and **reduction of financial benefit** [64–66] as causes for employees' stress on COVID-19.

**Conceptual framework.**  After reviewing the literature, a conceptual framework and hypotheses were created, as shown in **Fig 1**.

## Hypothesis statement

Hypothesis 1: There is a significant impact of Aggressive Cost Reduction strategies on Business Sustainability

Hypothesis 2: There is a significant impact of Workplace Environment Changes on Business Sustainability.

Hypothesis 3: Aggressive Cost Reduction Strategies significantly impact on Business Sustainability, mediating Employees' Stress on COVID-19.

Hypothesis 4: Workplace Environment Changes' impact on Business Sustainability significantly mediating Employees' Stress on COVID-19.

Hypothesis 5: There is a significant impact of Employees' Stress on COVID-19 on Business Sustainability.

## Material and methods

This is a cross-sectional study based on the positivism paradigm. Quantitative data have been collected using a deductive approach. The researchers applied cluster sampling methods to select the top 10 textiles and apparel manufacturing companies in Sri Lanka based on revenue

earned in August 2021 [67]. Out of 180,384 employees in these ten companies, 384 were selected for the study using random sampling [68].

Primary data were gathered using a structured bilingual questionnaire (i.e., Sinhala, English), as most employees were familiar with their mother tongue, Sinhala language. Due to the data collection issues encountered under pandemic circumstances, questionnaires were distributed via an online platform.

Multiple-choice questions were used to collect data on the demographic characteristics, and the five-point Likert scale was used to measure the independent, dependent, and mediating variables. The questionnaire consisted of five major sections. Section 'A'-demographic characteristics of respondents, Sections 'B' and 'C' on two independent variables, Section 'D' focuses on the mediating variable, and Section 'E' focuses on the dependent variable. The respondents were requested to rate on a scale of 1 to 5, where 1 = Strongly Disagreed to 5 = Strongly Agreed.

Collected data were appropriately analyzed and interpreted through descriptive statistics and SEM techniques. The SEM aims to expand the relationship between main variables and is a combination of two primary models: the measurement model (basically confirmatory factor analysis) and the SEM [69].

The SEM helps researchers examine complex relationships based on a structural model by representing the hypothesis between the dependent and independent variables. This model also predicts a series of interrelated dependence relationships simultaneously. This model is supported to provide a pictorial representation of the causal relationship among regression equations. The SEM is also a widely used statistical modeling tool that can be used to integrate factor analysis and regression path analysis.

Cronbach's alpha values were used to ensure reliability and test validity, whereas AVE was considered, and sample adequacy was used for factor analysis by KMO values on each item. Factor analysis was used to determine the grouping of correlations between latent variables and observable variables.

## Ethical consent

Researchers should be cognizant of ethical problems concerning the study. Accordingly, the study was reviewed and approved by the SLIIT Business School's ethical review committee. As part of research transparency, researchers were required to obtain participants' consent to take part in the study prior to its commencement. The participants were briefed about the study's nature, methodology, and research goal. The researcher has an ethical obligation to protect the participants' human dignity, privacy, and secrecy, as well as fairness and inclusivity. The purpose of the study and other details were duly disclosed to the relevant authorities and participants at the outset. All respondents were given the feel free to provide thoughts for responses to the questionnaire. Every respondent in this study confirmed their verbal consent before the data collection. Participants were not financially induced or coerced to take part in the study.

## Results and discussion

### Demographic characteristics

Table 1 shows the frequency and percentage results of each demographic factor.

The data collected using the questionnaire based on the respondent's demographic characteristics was presented and analyzed as gender, age, career level, salary, and year in service. According to the analyzed data, the number of female respondents waw 197 (51%) out of 384 total. Thus, most employees were female. Moreover, 148 (38%) of the employees were between

**Table 1. Demographic factors (N = 384).**

| Variable | Item | Frequency | Percentage |
|---|---|---|---|
| **Gender** | Female | 197 | 51.% |
| | Male | 187 | 49.% |
| **Age** | Below 20 years | 30 | 8% |
| | 21–30 years | 148 | 38% |
| | 31–40 years | 141 | 37% |
| | 41–55 years | 42 | 11% |
| | Above 55 years | 23 | 6% |
| **Career Level** | Managerial | 101 | 26% |
| | Non-Managerial | 283 | 74% |
| **Salary** | More than 70,000 | 55 | 14% |
| | 50,000–70,000 | 103 | 27% |
| | 25000–50,000 | 173 | 45% |
| | Below 25,000 | 53 | 14% |
| **Year in Service** | More than 25 years | 19 | 5% |
| | 10–25 years | 60 | 16% |
| | 5–9 years | 135 | 35% |
| | 1–4 years | 119 | 31% |
| | Below one year | 51 | 13% |

Source -Author's representation based on SPSS Results.

the ages of 21 to 30 years, showing that most were young. Their average (SD) age was 26.8 (9.73) years. The employees' median (Q1, Q3) age was 31–40 years (21–30, 31–40).

Moreover, 101 (26%) out of 384 employees were at the managerial level, and 283 (74%) were at the non-managerial level; thus, the majority were in the non-managerial group.

Meanwhile, 45% of employees earned a salary between Sri Lankan Rupees (Rs) 25,000 and 50,000 per month. Their median (Q1, Q3) salary level was Rs.50,000–70,000 (50,000–70,000, 25,000–50,000). Most employees (35%) have working experience of between 5 to 9 years. Only 19 (5%) employees have worked for over 25 years. In addition, 51(31%) employees had less than one year of experience. The descriptive statistics show the median (Q1, Q3) years in service was 5–9 years (5–9, 1–4). Therefore, it was assumed that most employees have had nearly five years of working experience in companies.

## Factor analysis and reliability and validity

Factor analysis is a technique that can be used to determine whether the grouping of correlations between latent variables and observable variables is justified. The researchers used factor analysis to examine the construction of variables or tests to decrease the number of theoretically relevant variables [70].

Table 2 presents each variable's Kaiser-Meyer-Olkin (KMO) values, which measured how suited data were for Factor Analysis. The test measured sampling adequacy for each variable in the model and the complete model. If KMO values are between 0.8 and 1, it indicates the sample is very well suited for factor analysis, and value between 0.7 to 0.8 is good enough to conduct a factor analysis [71]. The results show that the KMO values of the business sustainability variable were 0.788, with the aggressive cost reduction strategies variable with 0.774, Environmental workplace changes with 0.773, and employee stress with 0.849, which were good KMO values.

**Table 2. Factor analysis, validity, and reliability test.**

| Variable | | Factor Loading | KMO value | Cronbach's alpha value | AVE | CR |
|---|---|---|---|---|---|---|
| Business Sustainability | Q25_BS | 0.725 | 0.788 | 0.736 | 0.802 | 0.953 |
| | Q26_BS | 0.897 | | | | |
| | Q28_BS | 0.998 | | | | |
| | Q29_BS | 0.886 | | | | |
| | Q30_BS | 0.949 | | | | |
| Aggressive Cost Reduction | Q1_CR | 0.670 | 0.774 | 0.767 | 0.511 | 0.836 |
| | Q2_CR | 0.695 | | | | |
| | Q3_CR | 0.787 | | | | |
| | Q4_CR | 0.716 | | | | |
| | Q7_CR | 0.702 | | | | |
| Workplace Environmental Changes | Q10_WE | 0.723 | 0.773 | 0.772 | 0.587 | 0.850 |
| | Q11_WE | 0.768 | | | | |
| | Q12_WE | 0.779 | | | | |
| | Q13_WE | 0.793 | | | | |
| Employee Stress | Q14_ES | 0.663 | 0.849 | 0.830 | 0.509 | 0.860 |
| | Q15_ES | 0.679 | | | | |
| | Q16_ES | 0.607 | | | | |
| | Q17_ES | 0.763 | | | | |
| | Q20_ES | 0.742 | | | | |
| | Q24_ES | 0.806 | | | | |

Source: Authors' representation based on SPSS results.

Table 2 presents Cronbach's alpha and composite reliability values, which determine the validity and reliability of the study. According to the findings, Cronbach's alpha values show 0.736 for aggressive cost reduction strategies, 0.767 for workplace environmental changes, 0.772 for employee stress, and 0.830 for business sustainability, which depicted the internal reliability of the study. If Cronbach's alpha values are greater than 0.7, indicate the accepted internal reliability.

Convergent validity is best achieved if an Average Variance Extracted (AVE) is above 0.50 [72]. The study findings show that AVE values are 0.802 for aggressive cost reduction strategies, 0.511 for workplace environmental changes, 0.587 for employee stress, and 0.509 for business sustainability internal consistency of the present study. Composite Reliable (CR) values for all the variables above 0.8 confirmed reliability and internal consistency.

## Results of the structural equation model (SEM)

**Model fitting analysis.** The researchers examined the discrepancy between the model generated utilizing restricted independence or the null model, and no relationship existed between the observed variables to study the model fit. The Comparative Fix Index (CFI) can range from 0 to 1.0 but requires to be 0.9 or higher and less than 1.0 for an agreed-upon standard of a good model fit [73]. Further, RMSEA is a widely used model for testing research hypotheses by estimating the level of uncertainty to validate the goodness-of-fit. When selecting a good fit model, an absolute model fit value should be 0.08 or less. If the estimating process allows for higher value accuracy, the confidence interval should be about 90% [74].

All relevant index criteria values should meet the critical value boundaries to fit the proposed model, as shown in Table 3. Initially, the model used in the study did not derive the

**Table 3. Model fitting analysis.**

| Name of Category | Name of index | Model | Comment |
|---|---|---|---|
| Absolute fit | RMSEA | 0.066 | The required level is achieved |
| Incremental fit | CFI | 0.913 | The required level is achieved |
| Parsimonious fit | Chi-sq./df | 2.644 | The required level is achieved |

Source: Authors' representation based on AMOS results.

model fit and could not achieve the fitting index criteria as RMSEA = 0.080/ CFI = 0.794/ Chi-sq./df = 3.424 were not recorded as accepted level. Therefore CR5, CR6, WE8, WE9, ES19, ES21, ES22, and BS28 were removed considering the low factor loading items, and ES18 and ES23 were merged considering the modification indices. Afterward, the fitting index criteria values reached their required level (RMSEA = 0.066/CFI = 0.913/Chi-sq./df = 2.644) as shown in Table 3, and the initial model (**Fig 2**), and the model is good for further analysis.

## Regression results

Hypotheses one to five, $H_1$ to $H_5$, were developed to determine whether a significant relationship exists between business sustainability and aggressive cost reduction strategies ($H_1$), workplace environment changes ($H_2$), and employee stress on COVID-19 ($H_5$), and whether a significant relationship exists between employee stress on COVID-19 with aggressive cost reduction strategies ($H_4$), and workplace environment changes ($H_5$), respectively (**Fig 3**).

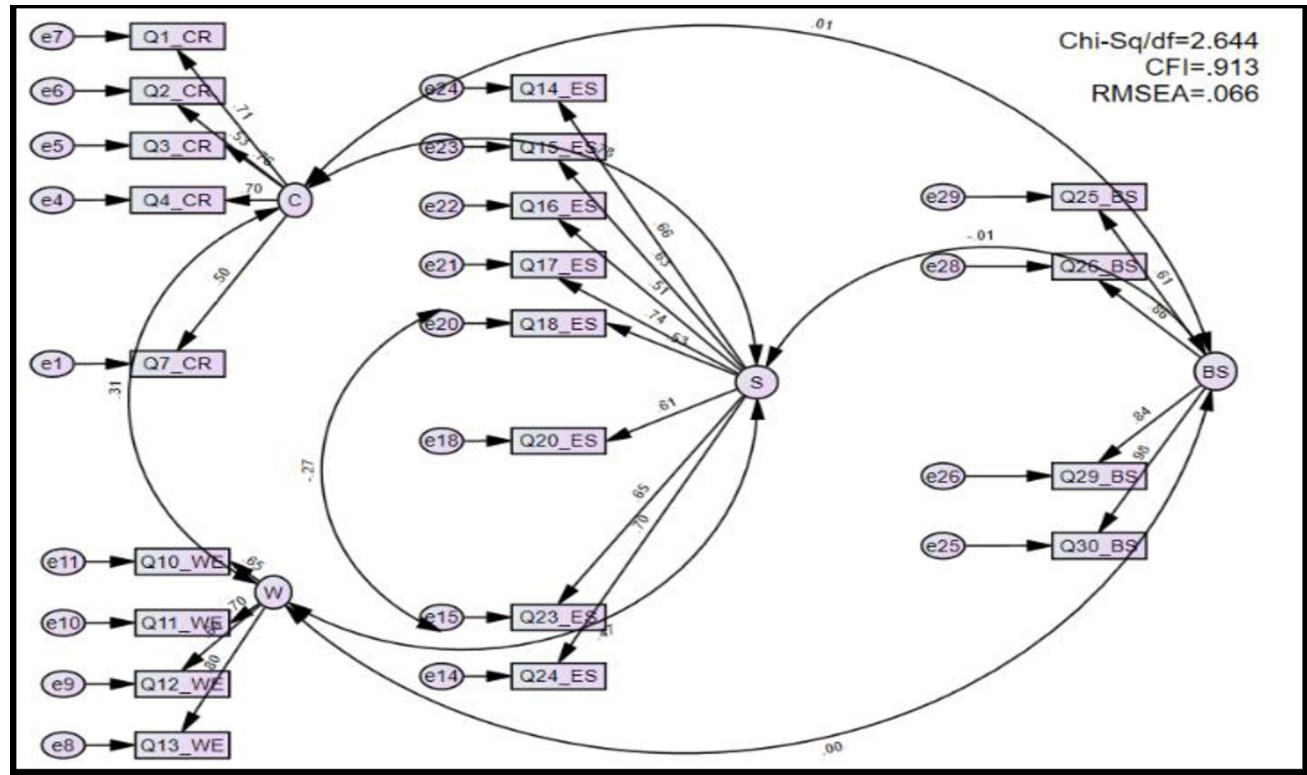

**Fig 2. Initial model.** Source: Author's illustration based on AMOS result.

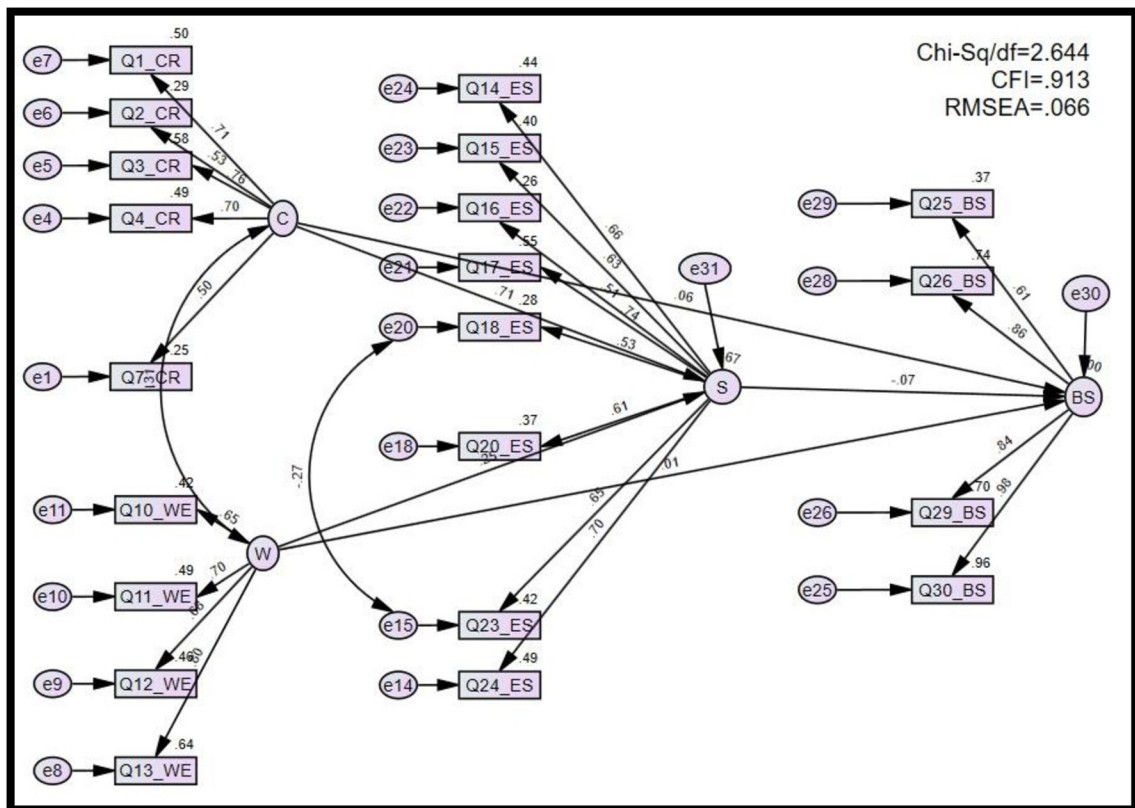

**Fig 3. Regression weights among variables.** Source: Author's illustration based on AMOS result.

The path analysis results, as shown in **Figs 3** and **4,** indicated that there is no relationship between business sustainability and aggressive cost reduction strategies ($\beta$ = .046, p < 0.608); business sustainability and workplace environment changes ($\beta$ = .006, p < 0.837) and business sustainability and employee stress on COVID-19 ($\beta$ = -0.028, p < 0.594) hence, $H_1$, $H_2$ and $H_5$ were rejected (Table 4). Thus, aggressive cost reduction strategies, workplace environment changes, and Employee stress on COVID-19 were not significant factors in business sustainability during COVID-19.

As shown in Table 4, there was a significant positive impact on employee stress on COVID-19 and aggressive cost reduction strategies ($\beta$ = 1.317, p < 0.001) and workplace environment changes ($\beta$ = .251, p < 0.001). Hence, $H_3$ and $H_4$ hypotheses were accepted. Employee stress on COVID-19 ($R^2$ = 0.67) was explained by 67% of these two variables (**Fig 4**). The results show that work aggressive cost reduction strategies and workplace environment changes could increase employee stress on COVID-19. The results of this study are similar to those of various previous studies.

## Discussion

The study's findings identified that aggressive cost reduction strategies had no significant impact on business sustainability. This scenario was because it was challenging for apparel companies to run a sustainable model at the initial pandemic stage. Companies were compelled to cut down salaries, bonuses, and employee layoffs as aggressive cost-cutting measures due to the profitability reduction. However, after several months, apparel players implemented new strategies to reduce aggressive costs, set stretch targets for the employees, and motivate

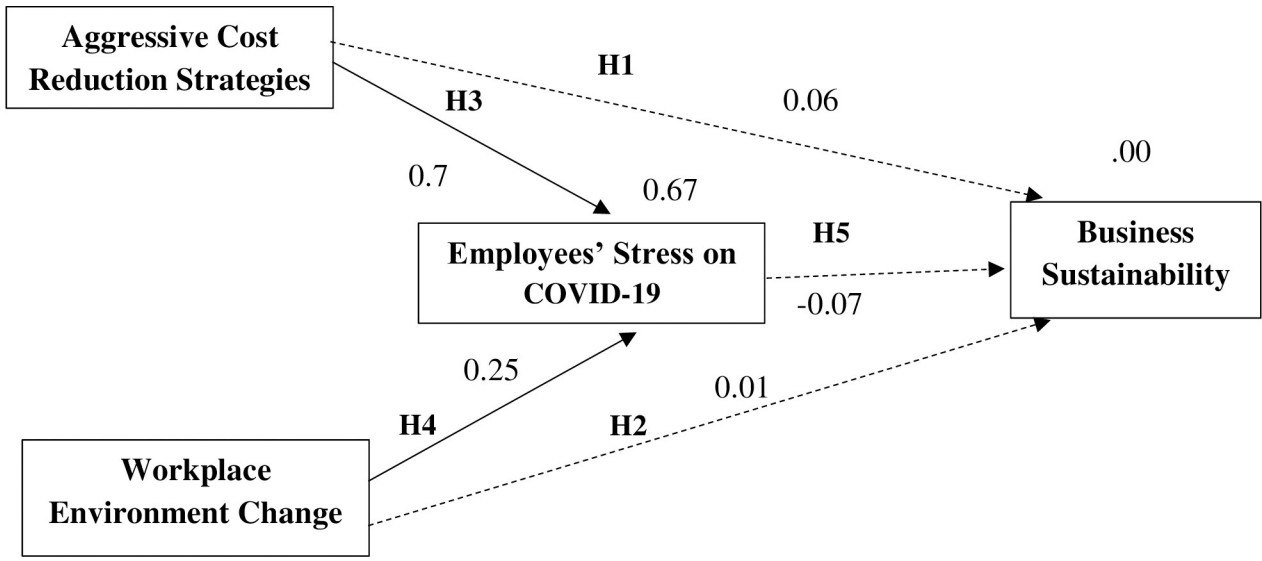

**Fig 4. Regression weight estimate values among variables.** Source: Author's illustration based on AMOS result.

them to work hard. Thereafter, a good level of supplier collaboration was noted. Furthermore, the government also supported sustaining the apparel industry during this pandemic. Accordingly, they have achieved breakeven profitability and aligned with the sustainable model. As such, these findings were supported in prior research [1].

The pandemic affected employee layoffs drastically in the apparel sector [21]. In addition to employee layoffs, the COVID-19 crisis has resulted in salary cuts/pay cuts, the elimination of various allowances, and the imposition of extended working hours and productivity targets by garment manufacturers [21]. Due to the critical situation, supply chains have been forced to cooperate with various stakeholders to minimize risk and uncertainty [38]. This collaboration has improved supply chain efficiency by lowering costs and increasing cash flow. Surplus inventory can result in waste simultaneously, as lacking inventory can lead to decreased lead time and poor customer experience. Order fulfillment/just-in-time delivery strategy implementation would have also minimized costs and helped supply chains survive [51].

The traditional working background has radically changed; hence, platform organizations encourage WFH, adaptable working times, and delocalized work to make earning subsisting uncomplicated for employees. Social obligations can be performed with flexible or restricted working hours [50].

**Table 4. Regression weight estimate values among variables.**

| | | | Estimate | P-Value | Results |
|---|---|---|---|---|---|
| Business sustainability | <— | Aggressive Cost reduction strategies | .046 | .608 | Reject H$_1$ |
| Business sustainability | <— | Workplace environment changes | .006 | .837 | Reject H$_2$ |
| Employee Stress on COVID-19 | <— | Aggressive Cost Reduction Strategies | 1.317 | *** | Accept H$_3$ |
| Employee Stress on COVID-19 | <— | Workplace environment changes | .251 | *** | Accept H$_4$ |
| Business sustainability | <— | Employee Stress on COVID-19 | -.028 | .594 | Reject H5 |

Note: S.E = Standard Error of estimate, C.R = Critical ration, P- Value = Significant value.

*** indicates highly significant at P<0.001.

Source: Authors' representation based on AMOS results.

The findings indicate that many apparel companies have moved to high-quality automated systems, because, due to higher absenteeism, employees were forced to share the workload of two employees with one worker. To meet the production goals on schedule and reduce export shipments, this was even more vital. With the WFH concept, the company can reduce both the environmental pollution and costs. A healthy and secure workplace helped them to maintain good productivity and employee satisfaction in the organization. Accordingly, the workplace environment also positively impacted the behavior of each employee. The management had set up a strategic plan (short term, long term) and motivated employees to reach targets. Because of these, the excellence in the working environment was a favorable setting that provided them considerable benefits for critical functions, such as improving employee satisfaction levels, productivity, profitability performance, and sustaining the business. These reasons indicate why workplace environmental changes did not significantly impact business sustainability in the apparel industry.

The findings show that during the COVID-19 crisis, employee stress was brought on by lay-offs, recruitment freezes, salary and bonus reductions, tighter waste management regulations, and increased stretch goals. This stress was brought on by job insecurity, diminished financial benefits, and a heavy workload.

As many business owners and companies have realized, employees have become one of a company's most valuable assets [24]. COVID-19's impact has resulted in new sources of financial, social, and physical stress far beyond the norm. According to a new MetLife mental health study, financial issues (81%) and job insecurity (77%) are some of the top stressors due to COVID-19 [54]. These stressors are long-lasting and widespread; if neglected, they would have threatened employee well-being [54]. Several apparel companies have been reported as having unpaid salaries during the pandemic [21].

According to a similar study, fear of getting infected (60%) and social distancing (47%) are some of the top stressors due to COVID-19 [54]. The affiliation to an organization affects how an employee behaves within an organizational setting, with employee motivation, inventiveness, absenteeism, fascination with fellow workers, and job retention. Nowadays, environmental changes are highly affected in many ways [75]. Seemingly, challenges resulting from prolonged social isolation can be costly and cause decreased productivity, increased levels of stress, adverse physical and mental conditions, and employee absence from being at duty or withdrawal from the team. The team's performance has been poor [76]. The previous study examined the effects of lockdown on employee mental health. It included the findings of employees who reported decreased motivation, loss of purpose and motivation, anxiety, and isolation due to mental health impairments [59].

However, in recent years, companies have been trying to implement strategies to manage absenteeism and employee turnover, which may have helped reduce employee stress. Also, the management motivates employees and continues to provide employee mental health awareness programs to reduce stress. E.g., WOW programs. Furthermore, the management of these companies has implemented new automation systems to lower employee pressure, thus achieving favorable results. Extending employee health and safety awareness campaigns and improving workplace health and safety has helped reduce employee stress due to life threats posed by COVID-19. The company has continued to prioritize maintaining employee morale during COVID-19, and management has offered mental and emotional support with a flexible working schedule during COVID-19. In doing so, the aim was to minimize employee stress to achieve better work performance, thereby upkeeping profitability and business sustainability.

However, according to the findings, during the COVID-19 crisis, multi-skilling, high workload, COVID-19-infected employees at the workplace, social isolation due to lockdown, and WFH were among the key reasons for the increased employee stress due to work environment changes.

The findings further revealed that stress in the apparel industry regarding health threats, lockdowns, social isolation and social distancing, job security, and a sharp drop in financial benefits have had no significant impact on business sustainability. Generally, the apparel industry provides and arranges employee motivation programs as well as improves training and development programs in respective organizations. As such, Sri Lanka's apparel industry is better poised to continue mental health awareness programs to reduce employee stress, implement new automation systems, and thereby lower employee pressure.

## Conclusions

The main objective of this research is to identify the factors that impact business sustainability in the apparel industry of Sri Lanka during the COVID-19 pandemic by exploring how aggressive cost reduction strategies, workplace environmental changes, and employee stress affect business sustainability. The results found that the majority of non-managerial (74%) employees are female (51%), with a 21–30-year range (38%) having a higher tendency to stress during pandemic situations. Most are in the 25,000–50,000-salary range (45%) with 5–9 years of service (35%) in the apparel sector. It can be concluded that aggressive cost reduction and environmental changes\ have a direct impact on employee stress rather than on business sustainability. Aggressive cost reduction and environmental changes account for a massive 67% of employee stress factors. To reduce employee stress, companies must eliminate factors that directly influence stress. As per the findings, eliminating cost reduction and environmental changes have a direct impact on eliminating employees' stress. Managers must focus on proposing and implementing policies for stress reduction. However, the findings revealed that cost reduction, environmental changes and employees stress do not directly or indirectly impact business sustainability. The results indicated that these three variables have little or no influence on business sustainability. Thus, the study recommends that policymakers identify real causes for business sustainability to minimize negative influences affecting business performance and survival during any crisis.

Business sustainability will probably stabilize the apparel industry and, in turn, will accelerate organizational performance and profitability. This research provides adequate evidence that aggressive cost reduction and environmental changes are essential in determining employee stress among operational-level employees in the apparel companies of Sri Lanka during COVID-19 but do not significantly affect business sustainability. The past literature supported that prior to the COVID-19, cost reduction through various strategies could enhance business sustainability [27]. However, the present study's findings revealed that neither cost reduction nor environmental changes or employee stress has effect on business sustainability during COVID-19. Thus, organizations should not employ these strategies to enhance business sustainability during crisis. The positive contribution of this study regarding critical analysis of aggressive strategies shows the significance of using aggressive strategies during crises to eliminate employee stress. Thus, it can be suggested that managers should make efforts to specifically target enhancing work satisfaction by eliminating factors that affect employee stress, such as aggressive cost reduction and environmental changes. Interventions of this type to combat such issues can effectively motivate operational-level employees, thereby retaining the best talent within the company for better organizational performance.

### Limitations and further study

The issue of aggressive cost cutting while maintaining business sustainability is relatively new. As such, currently, a limited number of studies are available with gaps in the literature. This study contributes to fill this research gap by focusing on the Sri Lankan apparel industry. It is

vital for apparel manufacturers to understand ways to reduce aggressive costs and control workplace environment changes, prioritizing workplace health and safety while maintaining business sustainability. The data were only limited to employees in the top Ten textiles and apparel manufacturing companies in Sri Lanka, and the sample size was 100 employees. The study's limitations include not promptly addressing issues of other severely impacted industries, such as tourism, education, transportation etc., and only reflecting the Sri Lankan apparel manufacturers' perspective in a situation where apparel manufacturers worldwide have been impacted. The authors recommend further research to investigate the COVID-19 impact on business sustainability by considering small and medium-sized apparel companies in Sri Lanka.

## Acknowledgments

The authors would like to thank Ms. Gayendri Karunarathne for language editing.

## Author Contributions

**Conceptualization:** Wasantha Rajapakshe, D. S. M. Karunaratna, W. H. G. Ariyaratne, H. A. Lakshani Madushika, G. S. K. Perera.

**Data curation:** Wasantha Rajapakshe, D. S. M. Karunaratna, W. H. G. Ariyaratne, H. A. Lakshani Madushika, G. S. K. Perera.

**Formal analysis:** Wasantha Rajapakshe, D. S. M. Karunaratna, W. H. G. Ariyaratne, H. A. Lakshani Madushika, G. S. K. Perera, P. Shamila.

**Methodology:** Wasantha Rajapakshe, D. S. M. Karunaratna, W. H. G. Ariyaratne, H. A. Lakshani Madushika, G. S. K. Perera.

**Project administration:** Wasantha Rajapakshe, P. Shamila.

**Resources:** W. H. G. Ariyaratne, H. A. Lakshani Madushika.

**Supervision:** Wasantha Rajapakshe, P. Shamila.

**Validation:** Wasantha Rajapakshe.

**Writing – original draft:** Wasantha Rajapakshe, D. S. M. Karunaratna, W. H. G. Ariyaratne, H. A. Lakshani Madushika, G. S. K. Perera, P. Shamila.

**Writing – review & editing:** Wasantha Rajapakshe.

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
