## [Decision Letter · Decision Letter 0]

3 Mar 2023

PONE-D-23-03480Do Aggressive Strategies Adopted during COVID-19 pandemic influence Business Sustainability in the Apparel Industry of Sri Lanka?PLOS ONE

Dear Dr. Rajapakshe,

Thank you for submitting your manuscript to PLOS ONE. After careful consideration, we feel that it has merit but does not fully meet PLOS ONE’s publication criteria as it currently stands. Therefore, we invite you to submit a revised version of the manuscript that addresses the points raised during the review process.

 The manuscript addresses an interesting research area but there is a requirement to incorporate changes suggested by the reviewers to make it suitable for the larger audience. 

We look forward to receiving your revised manuscript.

Kind regards,

Yasir Ahmad

Academic Editor

PLOS ONE

Additional Editor Comments: You should carefully read through the comments and address the observations. I would like to suggest modifying the title and write clearly the novelty of the research.  

Reviewers' comments:

Reviewer's Responses to Questions

**Comments to the Author**

1. Is the manuscript technically sound, and do the data support the conclusions?

Reviewer #1: Partly

Reviewer #2: No

Reviewer #3: Partly

2. Has the statistical analysis been performed appropriately and rigorously? 

Reviewer #1: Yes

Reviewer #2: Yes

Reviewer #3: Yes

3. Have the authors made all data underlying the findings in their manuscript fully available?

Reviewer #1: No

Reviewer #2: Yes

Reviewer #3: Yes

4. Is the manuscript presented in an intelligible fashion and written in standard English?

Reviewer #1: Yes

Reviewer #2: Yes

Reviewer #3: No

5. Review Comments to the Author

Reviewer #1: During the study author tested the hypotheses on 384 respondents by using the structured equation modeling with the aid of AMOS and concluded that aggressive cost reduction and environmental changes,

have a direct impact on employee stress rather than on business sustainability.

Please revise the title of the paper. Usually, it is not suggested to use the question in titles. Because the aim of conducting research is to find answers and having a question in your title may not be attractive to the readers.

Please revise the abstract and include the result of empirical findings in the abstract.

Line 56 I cannot cross-check the references. In addition, cited statistics are old, and please cite the up-to-date statistic for 2022.

Line 63, please revise the sentence “As countries…”

Line 67, Please specify earlier crises, such as....

Line 90, Please revise the paragraphs into the past tense as the pandemic is already over, and the world is moving forward. For instance, The Joint Apparel Association 90 Forum (JAAF) in Sri Lanka anticipated the epidemic.

Line 124- One of the study purposes is to identify the impact of cost-cutting strategies on apparel strategies during Covid. While in the introduction, I can not see any information about the cost-cutting strategies what tactics or strategies had been adopted by the company to reduce the cost, while I think all the implementation, such as health and safety measures and completion of orders during Covid also increased the price due to high carriage rates.

I think the author should move the first paragraph of the literature into the introduction section, Lines 132-140, or explain the business suitability in the introduction section as well.

In line 173- Just in Time delivery Strategy, I wonder that in the introduction section author stated that due to the breakage in the supply chain and working from home decreased production, and most of the operations were shut down during the pandemic. How in-time delivery would reduce the carriage cost was high worldwide due to breakage in the supply chain.

I think the author should move lines 319 – 325 into the methodology section.

Line 414: From this result, the author concluded that “many apparel companies have moved high-quality automated systems because sometimes they have to cover two employee workloads with one employee due to increased absenteeism and how it is the cost reduction strategy in the short term if companies are investing huge amount on the machinery.

Please justify why the author adopted Structural equation modeling (SEM)

Include the limitation and guidelines for further studies.

Reviewer #2: This study highlights the importance of investing in business sustainability, particularly in the apparel industry, in order to ensure survival in challenging situations such as the COVID-19 pandemic. The author notes that while there have been numerous studies on business sustainability, there is still a need for further investigation into how factors such as aggressive cost reduction, organizational environment change, and employee stress impact business sustainability.

The study conducted by the author involved 384 respondents (How this been determined, please show detail sample size calculation) in the Sri Lankan apparel industry and utilized structured equation modeling with the aid of AMOS to analyze the data. The key findings indicated that aggressive cost reduction and environmental changes have a direct impact on employee stress, rather than on business sustainability. The author suggests that policymakers should identify the real causes for business sustainability in order to minimize negative influences that could be affected by sustainability.

Overall, this study provides valuable insight into the importance of business sustainability in the apparel industry, particularly in the current pandemic situation, and highlights the need for further research in this area. The study's findings suggest that policymakers should take a comprehensive approach to business sustainability that takes into account a range of factors, including cost reduction, organizational environment change, and employee stress.

Provide sritics of the literature you reviewed. Also it it better to provide summary of the literature review and it needs to be reflected in the research gap or need of the study.

It is better to provide mean age (SD), in salary and year in service please show median ( Q1, Q3)

Revise Intext citation formatting.

Revise referencing formatting as per manuscript guideline again.

Reviewer #3: 1- the aim of study did not mentioned clearly in the abstract, please rewrite the abstract and mention the purpose of study, method, findings & conclusion briefly.

2- key words should be more relevant and correct based on MESH & COVID 19 should be added.

3- many sentences ( e.g. first and second sentences of second paragraph) do not have any references, please add

6. PLOS authors have the option to publish the peer review history of their article (what does this mean?). If published, this will include your full peer review and any attached files.

Reviewer #1: No

Reviewer #2: **Yes: **BIBHAV ADHIKARI

Reviewer #3: No

---

## [Author Response · Author response to Decision Letter 0]

15 Mar 2023

Date: Mar 14 2023 

To: "PLOS ONE" plosone@plos.org

From: "Wasantha Rajapakshe" wasantha.r@sliit.lk

Subject: PLOS ONE Decision: Revision required [PONE-D-23-03480]

Point by point response to editor and reviewers

Dear editor and the reviewers,

We would like to express our profound appreciation to the editor and the reviewers for the valuable comments and suggestions made on our manuscript which were very helpful in revising and improving it.

Please note that the line numbers referred to in this document is aligned with the file labeled 'Manuscript'.

Comments of Authors: This has been corrected in the revised version.

Comments of Authors: This has been corrected in the revised version. The revised version has been proofread. Corrections have been made including typos, comma correction and brevity.

Additional Editor Comments: You should carefully read through the comments and address the observations. I would like to suggest modifying the title and write clearly the novelty of the research. 

 Comments of Authors:

Noted with thank you. Title has been modified and novelty has been incorporated in the revised manuscript with track changes.

Reviewers' comments:

Reviewer's Responses to Questions

Comments to the Author

1. Is the manuscript technically sound, and do the data support the conclusions?

Reviewer #1: Partly

Reviewer #2: No

Reviewer #3: Partly

Comments of Authors: Noted with thank you. This has been incorporated in the revised manuscript with track changes. (Line 496-501)

“The results found that majority of non-managerial (73.7%) employees are female (51.3%), with 21-30 years range (38.7%) have higher tendency to stress during pandemic situation. Most of them are in the 20,000-50,000-salary range (45.1%) with 5-9 years of service (35.2%) in the apparel sector. In conclusion, it notes that aggressive cost reduction, as well as environmental changes, have a direct impact on employee stress rather than on business sustainability. To reduce employee stress, companies must eliminate factors that directly influence stress. As per the findings, eliminating cost reduction and environmental changes has a direct impact on eliminating employees’ stress. Managers must focus on proposing policies for stress reduction”. 

 2. Has the statistical analysis been performed appropriately and rigorously?

 Reviewer #1: Yes

Reviewer #2: Yes

Reviewer #3: Yes

Comments of Authors: Well, noted. Thank you for your comment.

 3. Have the authors made all data underlying the findings in their manuscript fully available?

 Reviewer #1: No

Reviewer #2: Yes

Reviewer #3: Yes

Comments of Authors:

Thank you comment has been noted. Data cannot be shared publicly because of data protection. Data are available from the Sri Lanka Institute of Information Technology, SLIIT Business School, Sri Lanka (contact via sandalimihara@gmail.com or wasantha.r@sliit.lk ) for researchers who meet the criteria for access to confidential data. However, for the review’s purposes, we are sending the data file as a supporting document.

 4. Is the manuscript presented in an intelligible fashion and written in standard English?

Reviewer #1: Yes

Reviewer #2: Yes

Reviewer #3: No

Comments of Authors:

Thank you and well noted. This has been in cooperated in the revised manuscript with track changes.

 5. Review Comments to the Author

Reviewer #1

Reviewer #1: During the study author tested the hypotheses on 384 respondents by using the structured equation modeling with the aid of AMOS and concluded that aggressive cost reduction and environmental changes, have a direct impact on employee stress rather than on business sustainability.

Please revise the title of the paper. Usually, it is not suggested to use the question in titles. Because the aim of conducting research is to find answers and having a question in your title may not be attractive to the readers.

Comments of Authors: Thank you and well noted. Title has changed to 

“Aggressive Strategies of the COVID-19 Pandemic on the Apparel Industry of Sri Lanka Using Structural Equation Modeling”

Please revise the abstract and include the result of empirical findings in the abstract.

Comments of Authors: Comment has been noted and this has been corrected in the revised manuscript. (Line 31-35)

“Aggressive cost reduction strategies (Beta = 1.274, p = 0.000) and environmental changes (Beta = 0.251, p = 0.000) led to an increase in employee stress but did not affect on business sustainability. Thus, employees stress (Beta = -0.028, p = 0.594) was not a mediator in the relationship between aggressive cost cutting strategies and business sustainability and business sustainability was not a dependent variable.”

Line 56 I cannot cross-check the references. In addition, cited statistics are old, and please cite the up-to-date statistic for 2022.

Comments of Authors:

Comment has been noted and this has been corrected in the revised manuscript in Line 69-71 with latest (2022) statistics with a new citation. 

“A study on waste generation in the garment industry in Sri Lanka proves that a considerable 39.4 % of the total wastes, followed by fabric and yarn is around 29.6% [10]”.

Line 63, please revise the sentence “As countries…”

Comments of Authors: Comment has been noted and this has been corrected in the revised manuscript (Line 78-80).

“International trade declined, supply chain operations were interrupted, and orders were canceled due to rapidly declining demand, showing the vulnerability of global supply chains to the COVID-19 pandemic situation.”.

Line 67, Please specify earlier crises, such as....

Comments of Authors: Comment has been noted and this has been corrected in the revised manuscript (Line 81-82).

“such as severe acute respiratory syndrome coronavirus (SARS-CoV) and Middle East respiratory syndrome coronavirus (MERS-CoV)”,

Line 90, Please revise the paragraphs into the past tense as the pandemic is already over, and the world is moving forward. For instance, The Joint Apparel Association 90 Forum (JAAF) in Sri Lanka anticipated the epidemic.

Comments of Authors: Comment has been noted and this has been corrected in the revised manuscript (Line 108-110).

“The Joint Apparel Association Forum (JAAF) in Sri Lanka anticipated that the pandemic affected 100,000 job losses in the apparel sector, with a 40% decrease in clothing and textile exports in the fiscal year 2020 [21]”.

Line 124- One of the study purposes is to identify the impact of cost-cutting strategies on apparel strategies during Covid. While in the introduction, I can not see any information about the cost-cutting strategies what tactics or strategies had been adopted by the company to reduce the cost, while I think all the implementation, such as health and safety measures and completion of orders during Covid also increased the price due to high carriage rates.

Comments of Authors: Thank you for your comment. This has been corrected in the revised manuscript (Page 4 - line 139-142).

“Many firms in sectors afflicted by the Covid-19 pandemic are deciding on salary cuts rather than layoffs to preserve their employees for a quick recovery [21]. Waste management, freezing recruitment and applying lean six sigma were basic tactics used by the firms to manage expenditure [7,11,19]. “ 

I think the author should move the first paragraph of the literature into the introduction section, Lines 132-140, or explain the business suitability in the introduction section as well.

Comments of Authors: Thank you for your comment. This has been corrected in the revised manuscript (Page 2 - line 44-52 and page 4 – Line 164-167).

“The management and coordination of environmental, social, and financial demands and concerns to achieve responsible, ethical, and long-term success are known as business sustainability [1]. Sustainable business strategies bring significant financial and environmental benefits to manufacturers, such as cost-effective procedures that reduce negative environmental consequences while preserving energy and natural resources. Employees, community, and product safety too are improved by sustainable production [2]. Three key components can be identified regarding business sustainability: employee satisfaction (people), long-term sustainable profit (profit), and a more sustainable planet with fewer resources polluting the environment (planet) [3].”

Line 164-167

“Sustainable manufacturing is defined as the production of manufactured goods using cost-effective procedures that reduce negative environmental consequences while preserving energy and natural resources. Employees, community, and product safety are all improved by sustainable production [2].” 

In line 173- Just in Time delivery Strategy, I wonder that in the introduction section author stated that due to the breakage in the supply chain and working from home decreased production, and most of the operations were shut down during the pandemic. How in-time delivery would reduce the carriage cost was high worldwide due to breakage in the supply chain.

Comments of Authors: Thank you for your comment and well noted. Even though there was interruption of supply chain, many researchers revealed that JIT was one way to apply warehouse cost reduction. Thus, with the literature support we think that also the possible strategy to reduce costs. However, we have adjusted the section as you mentioned. 

I think the author should move lines 319 – 325 into the methodology section.

Comments of Authors: Thank you for your comment. We have corrected this in the manuscript (Line 273-279).

“The SEM is also a widely used statistical modeling tool that can be used to integrate factor analysis and regression path analysis. This way, SEM helps researchers examine complex relationships based on a structural model by representing the hypothesis between the dependent and independent variables. According to [69], the SEM aims to expand the relationship between main variables and is surrounded by two primary models: the measurement model (basically confirmatory factor analysis) and the SEM.”

Line 414: Frlimitationssult, the author concluded that “many apparel companies have moved high-quality automated systems because sometimes they have to cover two employee workloads with one employee due to increased absenteeism and how it is the cost reduction strategy in the short term if companies are investing huge amount on the machinery.

Please justify why the author adopted Structural equation modeling (SEM)

Comments of Authors: Thank you for your comment. (Line 273-279)

“The SEM is also a widely used statistical modeling tool that can be used to integrate factor analysis and regression path analysis. This way, SEM helps researchers examine complex relationships based on a structural model by representing the hypothesis between the dependent and independent variables. According to [69], the SEM aims to expand the relationship between main variables and is surrounded by two primary models: the measurement model (basically confirmatory factor analysis) and the SEM.”

Include the limitations and guidelines for further studies.

Comments of Authors: Thank you for your comment. We have corrected this in the manuscript. (Line 518-530)

“The issue of aggressive cost cutting while maintaining business sustainability is relatively new. As such, currently, a limited number of studies are available with gaps in the literature. This study contributes to this gap and focused on the Sri Lankan apparel industry, which is vital for apparel manufacturers to understand ways to reduce aggressive costs and control workplace environment changes, prioritizing workplace health and safety while maintaining business sustainability. The data were only limited to employees in the top 10 textiles and apparel manufacturing companies in Sri Lanka, and the sample size was 100 employees. The study's limitations include not promptly addressing issues of other severely impacted industries, such as tourism, education, transportation etc., and only reflecting the Sri Lankan apparel manufacturers' perspective in a situation where apparel manufacturers worldwide have been impacted. The authors recommend further research to investigate the COVID-19 impact on business sustainability, by considering small and medium-sized apparel companies in Sri Lanka.”

Reviewer #2

Reviewer #2: This study highlights the importance of investing in business sustainability, particularly in the apparel industry, in order to ensure survival in challenging situations such as the COVID-19 pandemic. The author notes that while there have been numerous studies on business sustainability, there is still a need for further investigation into how factors such as aggressive cost reduction, organizational environment change, and employee stress impact business sustainability.

The study conducted by the author involved 384 respondents (How this been determined, please show detail sample size calculation) in the Sri Lankan apparel industry and utilized structured equation modeling with the aid of AMOS to analyze the data. The key findings indicated that aggressive cost reduction and environmental changes have a direct impact on employee stress, rather than on business sustainability. The author suggests that policymakers should identify the real causes for business sustainability in order to minimize negative influences that could be affected by sustainability.

Overall, this study provides valuable insight into the importance of business sustainability in the apparel industry, particularly in the current pandemic situation, and highlights the need for further research in this area. The study's findings suggest that policymakers should take a comprehensive approach to business sustainability that takes into account a range of factors, including cost reduction, organizational environment change, and employee stress.

Provide sritics of the literature you reviewed. Also it it better to provide summary of the literature review and it needs to be reflected in the research gap or need of the study.

It is better to provide mean age (SD), in salary and year in service please show median ( Q1, Q3)

Comments of Authors: Thank you for your comment. We have corrected this in the manuscript (Line 307-317).

“Their average (standard deviation) age was 26.8 (SD =9.73) years.

Moreover, 101 out of 384 were at the managerial level, and 283 were at the non-managerial level; thus, 73.7% of employees in the sample were in the non-managerial group.

Meanwhile, 45.1% of employees earned a salary between Sri Lankan Rupees (Rs) 25,000 and 50,000 per month. Their Median Salary Level Q1 was Rs.25,000- 50,000 while in Q3, it was Rs.50,000- 70,000. Most employees (35.2%) have had working experience of between 5 to 9 years. Only 19 employees have worked for more than 25 years. In addition to that, 51 employees were with less than one year of experience. The descriptive statistics show the Median year in service Q1 was 5–9 years while Q3 was 1-4 years.”

Revise Intext citation formatting.

Comments of Authors: Thank you for your comment. We have corrected this in the manuscript.

Revise referencing formatting as per manuscript guideline again.

Comments of Authors: Thank you for your comment. We have corrected this in the manuscript.

Reviewer #3

Reviewer #3: 1- the aim of study did not mentioned clearly in the abstract, please rewrite the abstract and mention the purpose of study, method, findings & conclusion briefly.

Comments of Authors: Thank you for your comment. We have corrected this in the manuscript. (Line 21-39)

“During the COVID-19 crisis, the apparel industry faced many challenges. Aggressive cost cutting strategies became a top priority, and in turn, these influenced stressors and adversely affected business sustainability. This study examines the impact of aggressive strategies of the COVID-19 pandemic on business sustainability in the apparel industry of Sri Lanka. Further, it investigates whether the relationship between aggressive cost cutting strategies and business sustainability was mediated by employee stress considering aggressive cost reduction strategies and workplace environmental changes. This was a cross sectional study with data collected from 384 employees in the apparel industry in Sri Lanka. Structural Equation Modelling (SEM) modeling was applied to analyze the direct and indirect effects of aggressive cost reduction strategies and workplace environmental changes on sustainability with mediating effects of stress. Aggressive cost reduction strategies (Beta = 1.274, p = 0.000) and environmental changes (Beta = 0.251, p = 0.000) led to an increase in employee stress but did not affect on business sustainability. Thus, employees stress (Beta = -0.028, p = 0.594) was not a mediator in the relationship between aggressive cost cutting strategies and business sustainability and business sustainability was not a dependent variable. The findings proved that managing workplace stress, particularly improving stressful working environments and aggressive cost reduction strategies can enhance employee satisfaction. Thus, it is plausible that managing employee stress could be benefit for policy makers to focus on which area they should identify to retain competent employees. “

2- key words should be more relevant and correct based on MESH & COVID 19 should be added. 

Comments of Authors: Thank you for your comment. We have corrected this in the manuscript. (Line 40-41)

Key Words: Business Sustainability, Aggressive Strategies, Employee Stress, Apparel Industry, COVID-19, Sri Lanka

3- many sentences ( e.g. first and second sentences of second paragraph) do not have any references, please add

Comments of Authors: Thank you for your comment. We have corrected this in the manuscript highlighted in red and with tract changes.

6. PLOS authors have the option to publish the peer review history of their article (what does this mean?). If published, this will include your full peer review and any attached files.

Comments of Authors: Yes

Do you want your identity to be public for this peer review? For information about this choice, including consent withdrawal, please see our Privacy Policy.

Reviewer #1: No

Reviewer #2: Yes: BIBHAV ADHIKARI

Reviewer #3: No

---

## [Decision Letter · Decision Letter 1]

16 Apr 2023

PONE-D-23-03480R1Aggressive Strategies of the COVID-19 Pandemic on the Apparel Industry of Sri Lanka Using Structural Equation ModelingPLOS ONE

Dear Dr. Rajapakshe,

Thank you for submitting your manuscript to PLOS ONE. After careful consideration, we feel that it has merit but does not fully meet PLOS ONE’s publication criteria as it currently stands. Therefore, we invite you to submit a revised version of the manuscript that addresses the points raised during the review process.

ACADEMIC EDITOR: The manuscript has improved a lot after the revision you submitted and the comments by the reviewers are quite encouraging. However, I still feel that with minor revision, your paper can gain more interest from the academic fraternity and provide valuable insights.I suggest to incorporate the following:Mention the contribution of the study in the abstract and introduction sections.Read carefully for the grammatical or punctuation errors ( I suggest getting help from the language expert)Be consistent with reporting the results of the study as mentioned by one of the reviewers.

We look forward to receiving your revised manuscript.

Kind regards,

Yasir Ahmad

Academic Editor

PLOS ONE

Journal Requirements:

Reviewers' comments:

Reviewer's Responses to Questions

**Comments to the Author**

1. If the authors have adequately addressed your comments raised in a previous round of review and you feel that this manuscript is now acceptable for publication, you may indicate that here to bypass the “Comments to the Author” section, enter your conflict of interest statement in the “Confidential to Editor” section, and submit your "Accept" recommendation.

Reviewer #1: All comments have been addressed

Reviewer #2: (No Response)

2. Is the manuscript technically sound, and do the data support the conclusions?

Reviewer #1: Yes

Reviewer #2: Partly

3. Has the statistical analysis been performed appropriately and rigorously? 

Reviewer #1: Yes

Reviewer #2: Yes

4. Have the authors made all data underlying the findings in their manuscript fully available?

Reviewer #1: Yes

Reviewer #2: Yes

5. Is the manuscript presented in an intelligible fashion and written in standard English?

Reviewer #1: Yes

Reviewer #2: No

6. Review Comments to the Author

Reviewer #1: Authors have addressed all the comments and in my opinion manuscript has been reached to the publishing quality. And I believe that the paper has improved significantly. I found the language of the text to be pleasant and easy to follow. I have the manuscript, which has already been revised and refers to the comments of the previous reviewers. Therefore, I have only a few more comments that the authors could address:

I would suggest the authors to emphasize more the scientific contribution of their research in the introduction.

Reviewer #2: Please revise the grammatical and punctuational errors. Technically paper is rigorious. however, "Their Median Salary Level Q1 was Rs.25,000- 50,000 while in Q3, it was Rs.50,000- 70,000. Most employees (35.2%) have had working experience of between 5 to 9 years. Only 19 employees have worked for more than 25 years. In addition to that, 51 employees were with less than one year of experience. The descriptive statistics show the Median year in service Q1 was 5–9 years while Q3

was 1-4 years." Looks fuzzy to me. Please mention them in format 'median(Q1,Q3)'.

7. PLOS authors have the option to publish the peer review history of their article (what does this mean?). If published, this will include your full peer review and any attached files.

Reviewer #1: No

Reviewer #2: **Yes: **BIBHAV ADHIKARI

---

## [Author Response · Author response to Decision Letter 1]

2 May 2023

Date: April 26, 2023

To: Prof. Yasir Ahmad

Academic Editor

PLOS ONE

From: Wasantha Rajapakshe

Subject: PONE-D-23-03480R1

Aggressive Strategies of the COVID-19 Pandemic on the Apparel Industry of Sri Lanka Using Structural Equation Modeling

Point by point response to editor and reviewers

Dear Respected Academic Editor and Reviewers,

We would like to express our profound appreciation to you for the valuable comments and suggestions made on our manuscript which were very helpful in revising and improving it.

Please note that the line numbers referred to in this document is aligned with the file labeled 'Manuscript'.

 Comments to the Author Comments of Authors

ACADEMIC EDITOR: 

The manuscript has improved a lot after the revision you submitted and the comments by the reviewers are quite encouraging. However, I still feel that with minor revision, your paper can gain more interest from the academic fraternity and provide valuable insights.

I suggest to incorporate the following: 

• Mention the contribution of the study in the abstract and introduction sections.

 Response to Editor: Thank you very much for your comments. This has been incorporated in the revised manuscript with track changes. (Line 41-44; 165-186)

• Read carefully for the grammatical or punctuation errors ( I suggest getting help from the language expert)

 Response to Editor: Thank you and well noted. This has been in cooperated in the revised manuscript with track changes.

• Be consistent with reporting the results of the study as mentioned by one of the reviewers.

 Response to Editor: Noted with thank you. This has been incorporated in the revised manuscript with track changes. (Line 524-529; 531-535; 540-550)

REVIEWERS:

1. If the authors have adequately addressed your comments raised in a previous round of review and you feel that this manuscript is now acceptable for publication, you may indicate that here to bypass the “Comments to the Author” section, enter your conflict of interest statement in the “Confidential to Editor” section, and submit your "Accept" recommendation. 

Reviewer #1: All comments have been addressed

 Response to Reviewer: Thank you very much for valuable time spend with our research and provide us valuable comments.

Reviewer #2: (No Response) 

Response to Reviewer: Thank you very much for valuable time spend with our research and provide us valuable comments.

2. Is the manuscript technically sound, and do the data support the conclusions?

Reviewer #1: Yes 

Response to Reviewer: Well, noted. Thank you for your comment.

Reviewer #2: Partly 

Response to Reviewer: Noted with thank you. This has been incorporated in the revised manuscript with track changes. (Line 524-529; 531-535; 540-550)

3. Has the statistical analysis been performed appropriately and rigorously? 

Reviewer #1: Yes 

Response to Reviewer: Well, noted. Thank you for your comment.

Reviewer #2: Yes 

Response to Reviewer: Well, noted. Thank you for your comment.

4. Have the authors made all data underlying the findings in their manuscript fully available?

Reviewer #1: Yes 

Response to Reviewer: Well, noted. Thank you for your comment.

Reviewer #2: Yes 

Response to Reviewer: Well, noted. Thank you for your comment.

5. Is the manuscript presented in an intelligible fashion and written in standard English?

Reviewer #1: Yes 

Response to Reviewer: Thank you very much for your comments.

Reviewer #2: No 

Response to Reviewer: Thank you and well noted. This has been in cooperated in the revised manuscript with track changes.

6. Review Comments to the Author

Reviewer #1: Authors have addressed all the comments and in my opinion manuscript has been reached to the publishing quality. And I believe that the paper has improved significantly. I found the language of the text to be pleasant and easy to follow. I have the manuscript, which has already been revised and refers to the comments of the previous reviewers. Therefore, I have only a few more comments that the authors could address:

I would suggest the authors to emphasize more the scientific contribution of their research in the introduction. 

Response to Reviewer: Thank you very much for your comments. This has been incorporated in the revised manuscript with track changes. (Line 41-44; 165-186)

Reviewer #2: Please revise the grammatical and punctuational errors. Technically paper is rigorous. however, "Their Median Salary Level Q1 was Rs.25,000- 50,000 while in Q3, it was Rs.50,000- 70,000. Most employees (35.2%) have had working experience of between 5 to 9 years. Only 19 employees have worked for more than 25 years. In addition to that, 51 employees were with less than one year of experience. The descriptive statistics show the Median year in service Q1 was 5–9 years while Q3

was 1-4 years." Looks fuzzy to me. Please mention them in format 'median(Q1,Q3)'. 

Response to Reviewer: Thank you very much for your comments. This has been incorporated in the revised manuscript with track changes. (Line 332-334; 338-339; 341-342)

7. PLOS authors have the option to publish the peer review history of their article (what does this mean?). If published, this will include your full peer review and any attached files.

Do you want your identity to be public for this peer review? For information about this choice, including consent withdrawal, please see our Privacy Policy.

Response to Reviewer: Yes. I would like to share peer review history.

Reviewer #1: No 

Reviewer #2: Yes: BIBHAV ADHIKARI

Thank you very much for your valuable comments.

Sincerely,

Wasantha Rajapakshe

(Corresponding Author)

---

## [Editor Report · Decision Letter 2]

22 May 2023

Aggressive Strategies of the COVID-19 Pandemic on the Apparel Industry of Sri Lanka Using Structural Equation Modeling

PONE-D-23-03480R2

Dear Dr. Rajapakshe,

We’re pleased to inform you that your manuscript has been judged scientifically suitable for publication and will be formally accepted for publication once it meets all outstanding technical requirements.

Kind regards,

Yasir Ahmad

Academic Editor

PLOS ONE
---

## [Editor Report · Acceptance letter]

25 May 2023

PONE-D-23-03480R2 

Aggressive Strategies of the COVID-19 Pandemic on the Apparel Industry of Sri Lanka Using Structural Equation Modeling 

Dear Dr. Rajapakshe:

I'm pleased to inform you that your manuscript has been deemed suitable for publication in PLOS ONE. Congratulations! Your manuscript is now with our production department. 

Kind regards, 

on behalf of

Dr. Yasir Ahmad 

Academic Editor

PLOS ONE